# Structural basis of translation inhibition by a valine tRNA-derived fragment

Yun Wu[1], Meng-Ting Ni[1], Ying-Hui Wang[1], Chen Wang[2,3], Hai Hou[4], Xing Zhang[2,3] , Jie Zhou[1]

**Translational regulation by non-coding RNAs is a mechanism commonly used by cells to fine-tune gene expression. A fragment derived from an archaeal valine tRNA (Val-tRF) has been previously identified to bind the small subunit of the ribosome and inhibit translation in *Haloferax volcanii*. Here, we present three cryo-electron microscopy structures of Val-tRF bound to the small subunit of *Sulfolobus acidocaldarius* ribosomes at resolutions between 4.02 and 4.53 Å. Within these complexes, Val-tRF was observed to bind to conserved RNA-interacting sites, including the ribosomal decoding center. The binding of Val-tRF destabilizes helices h24, h44, and h45 and the anti-Shine–Dalgarno sequence of 16S rRNA. The binding position of this molecule partially overlaps with the translation initiation factor aIF1A and occludes the mRNA P-site codon. Moreover, we found that the binding of Val-tRF is associated with steric hindrance of the H69 base of 23S rRNA in the large ribosome subunit, thereby preventing 70S assembly. Our data exemplify how tRNA-derived fragments bind to ribosomes and provide new insights into the mechanisms underlying translation inhibition by Val-tRFs.**

## Introduction

Non-coding RNAs (ncRNAs) are DNA-transcribed RNA molecules that, instead of encoding proteins, perform a range of important cellular functions that include the regulation of gene expression, control of cell differentiation, and maintenance of genome stability (1, 2, 3, 4, 5, 6, 7). Transfer RNA (tRNA) molecules play key roles in decoding the genetic information stored in mRNA and ensuring the accurate incorporation of amino acids into growing polypeptide chains during ribosomal translation (8, 9, 10, 11, 12, 13). tRNA-derived fragments (tRFs) are ncRNAs derived from tRNA, which have been found to play regulatory roles in a range of cellular processes (14, 15, 16, 17, 18) and have also been demonstrated to regulate protein translation (19, 20, 21, 22). The biological roles of these tRFs are mediated via a range of mechanisms through their interactions

with proteins or mRNAs to inhibit or promote translation. For example, with respect to the negative regulation of translation, the 5' fragments of tRNA[Ala] and tRNA[Cys] have been shown to inhibit global translation and promote stress granule formation by interacting with the YBX1 protein (23). Conversely, in the positive regulation of translation, a 22-nt tRF from tRNA[Leu] has been found to promote ribosome biogenesis by base-pairing with ribosomal protein mRNAs to enhance their translation (24). Our current structural understanding regarding the binding of ncRNAs to ribosomes (but not tRNA or mRNA) is based on several structures of internal ribosome entry site ribosome or transfer-messenger RNA (tmRNA)·ribosome complexes (25, 26, 27, 28, 29, 30, 31). These studies have enabled an elucidation of how tmRNAs regulate translational elongation and how the internal ribosome entry sites mediate the initiation of translation. Accordingly, studying the binding of tRFs to ribosomes can contribute to enhancing our mechanistic understanding of the translational regulation mediated by ncRNAs.

Valine tRNA (Val-tRF) is a 26-nucleotide-long fragment derived from the 5' region of the Val-tRF in the halophilic archaeon *Haloferax volcanii* (32), which is produced primarily in response to alkaline stress (unifying nomenclature: tDF-1:26-Val-GAC-1). Val-tRFs have been shown to interact with components of the translation machinery, including 30S or 70S ribosomes, to inhibit translation by preventing their association with mRNAs, thereby inhibiting the initiation of protein synthesis (33). Further evidence has indicated that Val-tRF can inhibit peptide bond formation, and consequently, it may also interfere with protein elongation. Moreover, it has been established that Val-tRF can attenuate protein translation in vivo and in vitro in both eukaryotic (*Saccharomyces cerevisiae*) and bacterial (*Escherichia coli*) systems (33). It can thus be assumed that the binding of Val-tRF and the mechanisms associated with its inhibition of protein translation have been evolutionarily conserved.

In this study, we investigated the structure of Val-tRF bound to an archaeal ribosomal 30S small subunit (SSU). Val-tRF was found to bind to the 30S SSU in three distinct states, characterized by differences in the interactions with ribosomes and the conformation of Val-tRF. Structural analysis further revealed the potential

---

[1]Life Sciences Institute, Zhejiang University, Hangzhou, China   [2]Center for Cryo-Electron Microscopy, Zhejiang University School of Medicine, Hangzhou, China [3]Department of Pathology of Sir Run Run Shaw Hospital and Department of Biophysics, Zhejiang University School of Medicine, Hangzhou, China   [4]Institute of Medical Research, Northwestern Polytechnical University, Xi'an Shaanxi, China

Correspondence: xzhang1999@zju.edu.cn; jiezh@zju.edu.cn

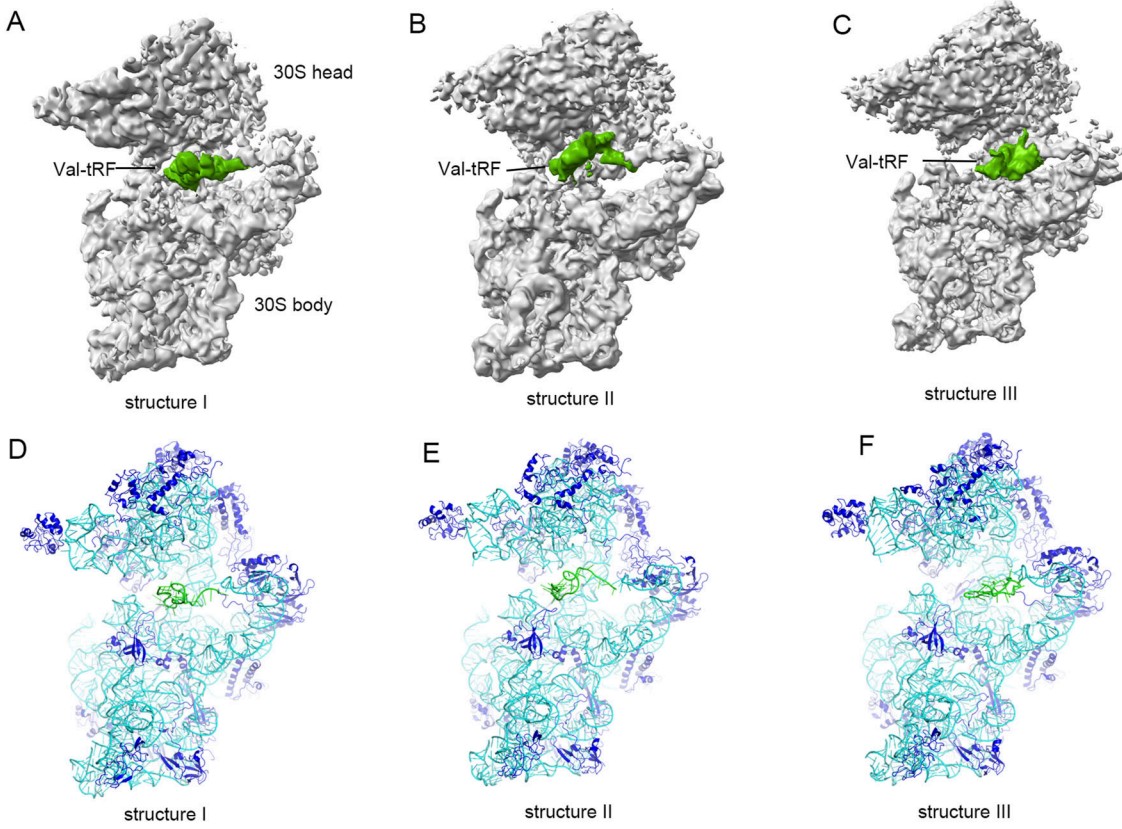

**Figure 1. Cryo-EM maps and the overall structures of the complexes between a valine tRNA-derived fragment (Val-tRF) and ribosomal RNA.**
**(A, B, C)** EM density maps for structures I (A), II (B), and III (C). **(D, E, F)** Refined atomic models of structures I (D), II (E), and III (F). For the EM maps: 30S, gray; Val-tRF, green. For the models: 16S rRNA, cyan; r-proteins, blue; Val-tRF, green.

influence of ribosome association and tRNA and mRNA binding in response to the binding of Val-tRF to the 30S SSU. These structures provide new insights into the mechanisms underlying the translational regulation mediated by tRNA-derived ncRNAs.

# Results

### Structural determination

The findings of previous studies on archaeal ribosomes derived from *Sulfolobus acidocaldarius* (*Sac*) have indicated their good stability and hence their suitability for structural studies (34). In the present study, we used *Sac* ribosomes to investigate the structure of the 30S–Val-tRF complex, as we speculated that this complex would have a stable conformation, particularly in the 30S head domain.

The in vitro-transcribed Val-tRF (bases 1–26 from the 5′ end of val-tRNA) was incubated with the *Sac* 30S SSU, and this sample was subjected to cryo-EM data collection. Given the presence of particle orientation preferences in this sample, we collected tilted data. This strategy enables us to provide additional information regarding the orientation of individual particles for 3D reconstruction, although it has the downside of reducing the overall resolution of the structure (35). Initial reconstruction and refinement enabled us to obtain a

high resolution for the 30S body/platform domain, whereas the densities for the 30S head domain and Val-tRFs were relatively poor (Figs S1, S2, and S3). We suspect that this contrast in resolution could be attributable to the conformational heterogeneity of the head domain structure. We initially performed extensive 3D classification of the 30S head domain, followed by a classification of Val-tRFs with a fixed 30S head orientation (36) (Figs S1A–C and S2A–C). Before classification, we observed that the density of Val-tRFs was located between the 30S SSU platform and shoulder (Fig S3A–E). A mask encompassing the Val-tRF density was used for focused 3D classification of the ligand region, which enabled us to solve a number of 30S–Val-tRF complex structures with different 30S heads and Val-tRF configurations. Indeed, we identified ~30 different Val-tRF conformations (Fig S4A–L). This can be attributed to the fact that the stability of the overall structure and conformation of Val-tRF is ensured by just two base pairs within Val-tRF. Meanwhile, the rest of the single-stranded RNA enhances its flexibility. However, the elements facilitating binding to the 30S SSU remained relatively stable (discussed below). Using this classification procedure, we selected three representative classes based on high particle abundance. The clear secondary structure from the EM maps enabled us to perform model construction and refinement. We eventually solved structures I, II, and III at respective resolutions of 4.10, 4.53, and 4.02 Å (comprising 9,472, 5,281, and 6,795 particles, respectively) (Figs 1A–F and 2A–C, Table S1). However, the

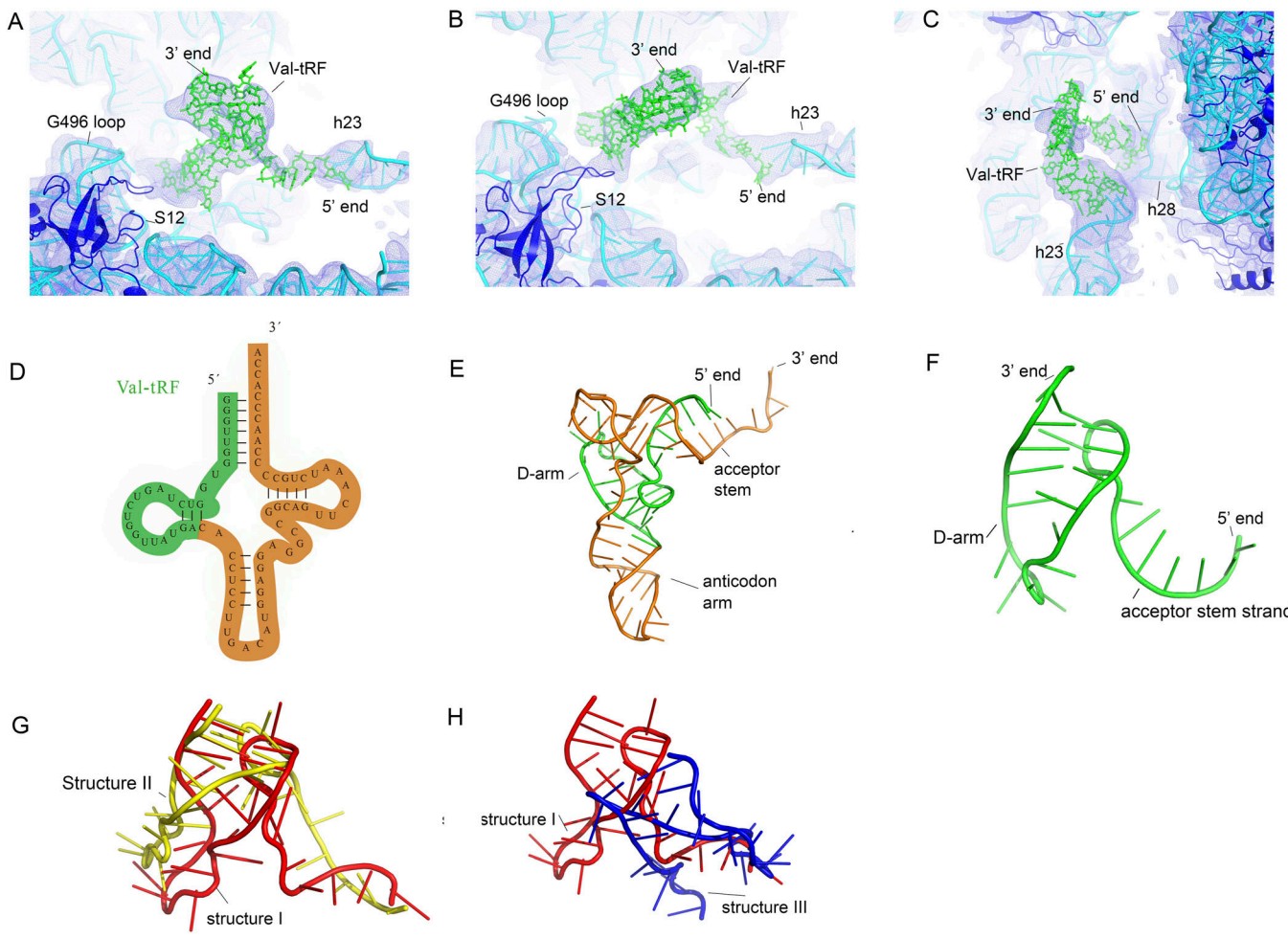

**Figure 2.  Cryo-EM maps and conformations of a valine tRNA-derived fragment (Val-tRF).**
**(A, B, C)** EM density map surrounding Val-tRF and the neighbour 16S rRNA helices in structures I (A), II (B), and III (C). **(D, E)** The Val-tRF sequence (green) within the parent Val-tRNA (orange and green). **(F)** The Val-tRF structure derived from val-tRNA structure (PDB: 4W29). **(G)** Conformational differences between Val-tRF from structure I (red) and structure II (yellow). **(H)** Structural differences of Val-tRF in structures I (red) and III (blue).

resolution of these three structures was limited to a certain extent by the relatively small number of particles.

**Conformations of Val-tRF**

The Val-tRF derived from Val-tRNA consists of a single strand of the tRNA acceptor stem (residues 1–7) and the D-arm (residues 5–26). The D-arms of tRNAs contribute to stabilizing the overall structure (Fig 2D–F). During the process of tRNA–ribosome binding, the D-arm is involved in the recognition and binding of tRNA to ribosomal A, P, and E sites (37, 38), whereas the tRNA acceptor stem plays a pivotal role in the charging and delivery of amino acids during protein synthesis. During this binding, the acceptor stem and D-arm primarily interact with the large ribosomal subunit. Moreover, the conformation of these two tRNA elements is relatively stable during tRNA movement from the A-site to the P-site and to the E-site. Previously obtained high-resolution structures of ribosome complexes bound to tRNAs have revealed that tRNAs can undergo large conformational changes

when they bind to different sites or when they occur in different functional states (such as hybrid or chimeric states) (39, 40, 41, 42, 43, 44). As a component of intact tRNA, Val-tRF binds to the ribosomal 30S SSU, and we found that its structural conformation may differ, which alters its interactions with the 30S SSU.

Having aligned the D-arm domain of Val-tRF, we noted that the single-stranded acceptor stems swing relative to each other (Fig 2G and H). Specifically, even though the binding interactions between Val-tRF with structures I and II and 30S were observed to be similar, the single-strand acceptor stem differed by ~120°. The D-arm domain is an RNA stem-loop structure comprising two base pairs, and although we established that the interactions between the two base pairs are maintained within structures I–III, the loop region undergoes conformational changes that are induced by distinct contacts with the 30S SSU. Contrary to the uniform folding pattern observed in the binding of translation factors to ribosomes (for instance, the initiation factor aIF1 binding to the 30S A site), Val-tRF exhibits various configurations, manifesting in altered binding patterns. This variance is believed to mirror

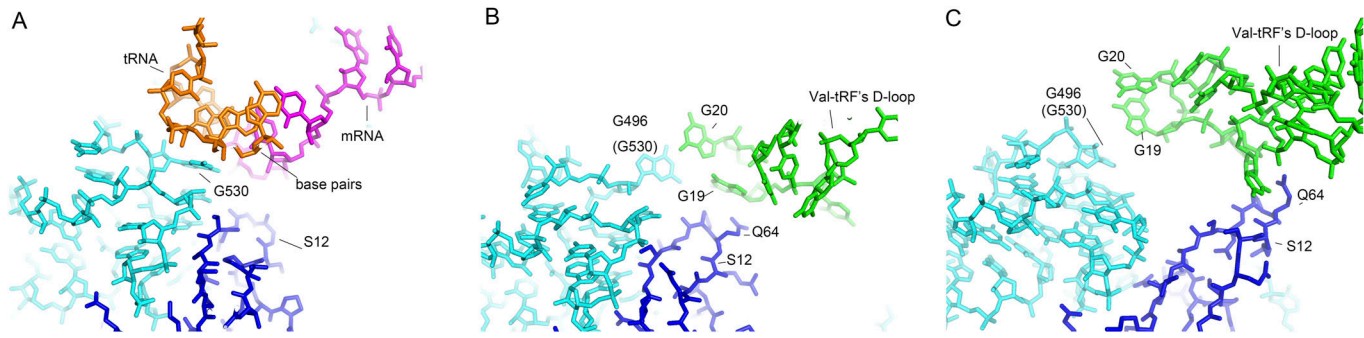

**Figure 3. The D-stem-loop of a valine tRNA-derived fragment (Val-tRF) that mimics the tRNA anticodon stem-loop.**
**(A)** The G530 loop interacts with the minor groove of the codon–anticodon helix in the structure of the 70S ribosome–tRNA–mRNA complex. **(B)** In structure I, bases G19 and G20 in the D-loop of Val-tRF lie very close to G496 (G530 in *Escherichia coli*) and Q64 of the uS12 protein and plausibly undergo interactions with these residues. **(C)** Similarly, bases G19 and G20 in Val-tRF come into contact with the G496 loop of 16S rRNA. In addition, C17 of Val-tRF may establish contact with Q64 of uS12.

differences in their chemical compositions and structural characteristics. The amino acids that make up proteins possess side chains capable of engaging in diverse interactions, including hydrogen bonding, electrostatic interactions, and hydrophobic interactions. These interactions allow proteins to assume relatively stable three-dimensional structures. In contrast, Val-tRF primarily relies on two complementary base pairs to maintain its structure and folding. This accounts for the conformational flexibility of Val-tRFs on the 30S ribosome.

### Val-tRF D stem-loop contacts with the decoding center

In structures I and II, we observed that the D-loop of the Val-tRF interacted with the G496 loop (G530 in *E. coli*) of the 16S rRNA and the uS12 protein (Fig 3A–C). The resolution of the cryo-EM maps enabled us to determine the potential residue interactions (Fig 2A–C). In structure I, G496 establishes contact with G19 and G20 via base interactions. The side chain of Q64 in u12 interacts with G19 ribose. These interactions are reminiscent of decoding center contacts during A-site tRNA anticodon stem-loop (ASL) binding. During translation, the binding of the tRNA molecule to the A site of ribosomes involves the recognition of the anticodon on the tRNA and the codon on the mRNA. This interaction induces a structural change in the 30S subunit that causes the decoding center (key residues include A1492, A1493, G530 of 16S rRNA and the side chains of uS12), which is located in the SSU of the ribosome, thereby adopting a conformation that is complementary to the codon-anticodon base pair (Fig 3A) (12, 45, 46). In structures II and III, A1503 and A1504 (A1492 and A1493 in *E. coli*) in helix h44 were incompatible with Val-tRF and became flexible. However, the D-loop of Val-tRF in these structures lies adjacent to the G496 loop and S12. These interactions indicate that the 30S SSU adopts similar mechanisms for recognizing and binding the RNA stem-loop.

### The acceptor stem strand of Val-tRF establishes contact with the h23 stem-loop of 16S rRNA

On inspecting the eight-nucleotide-long strand of Val-tRF derived from the acceptor arm in structures I and II, we observed

interactions between this strand and the h23 stem-loop (Fig 2A and B), which was found to interact with E-site tRNA ASL in bacterial ribosomes (41, 42, 43, 47, 48) (Fig S5). The structural model refined with reference to the EM maps indicated possible interactions between Val-tRF and helix h23. In these two structures, the G659 and A660 bases of the 16S rRNA were observed to be in contact with phosphates of the G2 of Val-tRF. In addition, base A661 of the 16S rRNA was found to stack with the G1 of Val-tRF. The interaction between h23 and the ASL of tRNA at site E of the bacterial ribosome is believed to be important for stabilizing the binding of tRNA to the ribosome and facilitating the release of spent tRNAs from the E site. In structures I and II, the interactions between Val-tRF and the h23 stem-loop indicate that h23 may be a target of RNA fragment binding.

### Structure III reveals a different binding pattern of Val-tRF

In contrast to structures I and II, in which Val-tRF bridges the G496 loop (G530 in *E. coli*) in the decoding center and the h23 helix, in structure III, we observed a 5 Å movement of Val-tRF, which resulted in a new binding site for this molecule (Fig 1C and F). Instead of coming into contacting with the G496 loop, the D loop of Val-tRF forms a network of interactions with the h23 stem-loop, whereas the acceptor strand establishes contact with the h28 of 16S rRNA. In addition, the D loop of Val-tRF established a close proximity to the h23 stem-loop, with which it interacts, plausibly by initiating several stacking interactions; however, the resolution was insufficiently clear to ascertain the details. These interactions may contribute to destabilizing the interactions between helices h23 and h24, thereby rendering h24 more flexible (Figs 4 and S6A and B). In this regard, it is noteworthy that the h24 stem-loop is also disordered in structures I and II, which may be attributable to the interactions between Val-tRF and h23, as mentioned above. In the 30S ribosomal subunit, h24 lies adjacent to h23, and its conformation is stabilized by bases at the tip of h23. One of the important functions of h24 in 16S rRNA is to establish contact with the P-site of the tRNA (such as fMet-tRNA) anticodon stem-loop during binding (41, 42, 43, 47, 48). Thus, a destabilization of h24 may explain the function of Val-tRFs in preventing initiator tRNA binding and inhibiting the initiation of translation.

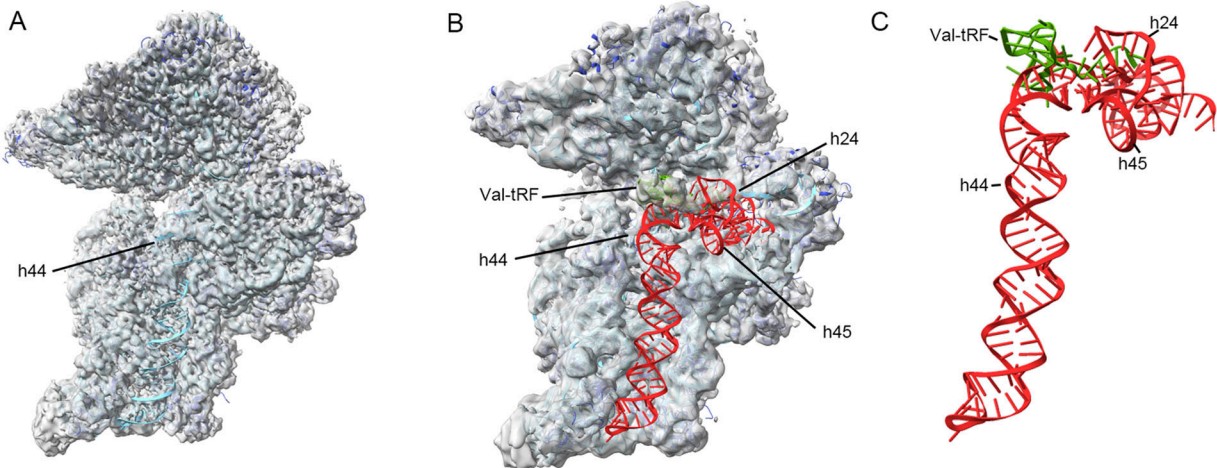

**Figure 4. Binding of a valine tRNA-derived fragment (Val-tRF) with the 30S ribosomal small subunit destabilizes helices h24, h44, and h45.**
**(A)** A cryo-EM map of the *Sulfolobus acidocaldarius* 30S small subunit without ligand binding (EMD-34862). **(B)** In structure I, binding of Val-tRF results in a lack of density for h24, h44, and h45. **(C)** The binding site of Val-tRF overlaps with the three helices of 16S rRNA, which may possibly account for the lack of density in the structures of these helices.

h28 is a 16S rRNA helix that bridges rRNA in the ribosomal body and head domains (49). Several structures have shown that translation initiation requires a large-scale rotation of the 30S head domain (50, 51, 52). The head rotates by pivoting around the neck helix (h28) of the 16S rRNA (49), which is the sole covalent connection to the body domain. h28 is characterized by a region lacking stringent base pairing. Previous structures have established that G896 flips from this helix and that G895 does not pair with the corresponding base. The acceptor strand of Val-tRF comes into contact with these two bases via G2 and G3 by establishing network interactions with the backbones of G895 and G896 (Fig S6C and D). It is speculated that the interactions between the 30S ribosome neck and Val-tRNA may influence the dynamic movements of the head domain, which are required for the translation initiation process.

### Binding of Val-tRF destabilizes the h44, h45, and h24 helices and Shine–Dalgarno sequence of 16S rRNA

In all three complex structures, we identified several 16S rRNA helices, in which the anti-Shine–Dalgarno (SD) strands became flexible, and no cryo-EM densities were attributed to these components (Fig 4). During structural model construction, we initially found that h44 clashes with the Val-tRF models (Fig S7A). Steric incompatibility results in an unstable conformation of h44, and thus, no h44 density was observed in the EM map. In contrast, in the EM map of the *Sac* 30S ribosome without ligand binding, we were able to trace a portion of the unstable h44 (Fig 4A), which can significantly influence ribosomal translation. h44 is a binding target for multiple translation initiation factors (IFs), including aIF1, aIF1A, and aIF2 (53, 54, 55), a group of proteins that play key roles in the initiation of protein synthesis. Binding defects in these IFs directly influence ribosome assembly on the mRNA molecule and the positioning of the initiator methionine-tRNA (tRNA$^{fMet}$) at the correct location on the ribosome. Furthermore, h44 is the longest helix of the 16S rRNA that spans the 30S interface from the neck region to the spur and contributes to multiple bridge

interactions, including B2a, B3, B5, and B6 (34, 37, 38, 41, 43). The flexible conformation of h44 conformation can directly influence the association of the 30S SSU with the 50S large subunit, which is essential for subsequent translational elongation.

h44 is linked to h45 and the downstream anti-SD sequence via a single-stranded RNA fragment. Modeling of Val-tRFs also revealed clashes between this RNA fragment and Val-tRFs, thereby accounting for the absence of the densities of h45 and anti-SD sequences (Figs 4B and C and S7A). h45 is a stem-loop structure that is among the most conserved helices in 16S rRNA and plays important roles in 70S ribosome assembly and the initiation of protein synthesis (56, 57). Destabilization of h45 may indicate that its functional deficiency was disrupted by Val-tRF. The anti-SD sequence interacts with the SD sequence upstream of the initiation codon (AUG) of the mRNA and thereby facilitates association of the mRNA with the ribosomal SSU. The anti-SD sequence acts as a competitor for the true SD sequence, thereby preventing the non-specific binding to ribosomes and ensuring that the initiation of translation occurs only at the correct start codon. This specificity accordingly contributes to maintaining the fidelity of protein synthesis. A flexible anti-SD sequence affects base-pairing interactions with the SD sequence of the mRNA, and thus for those mRNAs containing an SD sequence, the correct positioning of the start codon can be affected.

With respect to h24, we observed an absence of EM density, not only in structure III but also in structures I and II (Fig S7B), in which the contacts between the acceptor strand of Val-tRNA and h24 induced a 4.5 Å displacement of h24, thereby disrupting the contacts between h24 and h23, which play an important role in stabilizing the conformation of h24. Indeed, the visible portion of h24 already showed an ~6° movement induced by this instability.

### Exclusion of the mRNA P-site codon by Val-tRF

During the initiation of translation, mRNAs recruit the 30S ribosome and position the AUG initiation codon at the P-site. In archaea, this

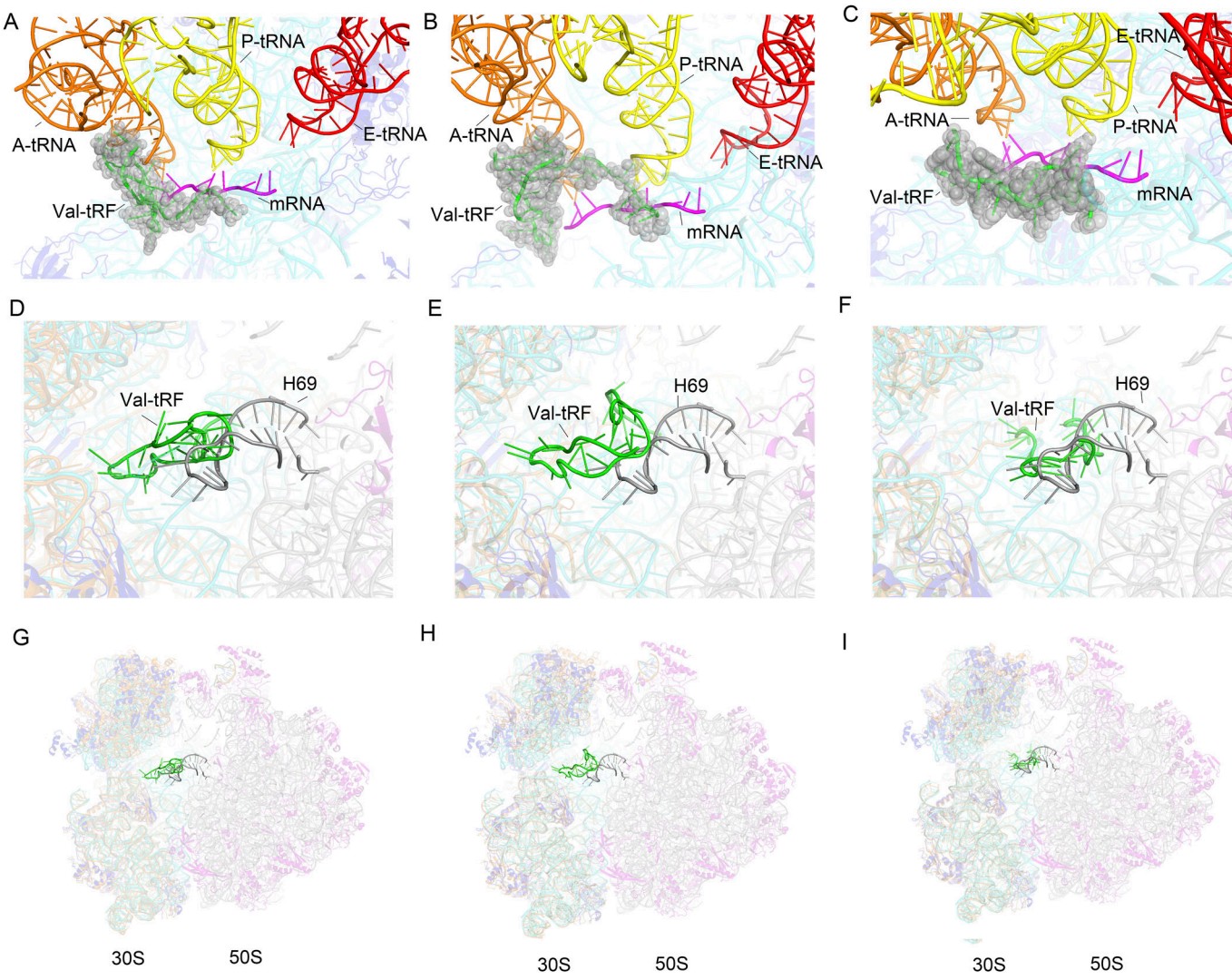

**Figure 5. P-site codon exclusion and inhibition of 70S association by a valine tRNA-derived fragment (Val-tRF).**
**(A, B)** In structure I or II, the 5' accepter-stem strand of Val-tRF clashes with the mRNA P-site codon if it binds. In addition, the D-stem-loop clashes with classical state A-site tRNA. **(C)** In structure II, the 5' strand of Val-tRF clashes with mRNA P-site codon. **(D, E)** Docking of the 30S–Val-tRF complex structure onto the *Sulfolobus acidocaldarius* 70S ribosome showing that Val-tRF would clash with H69 of the 23S rRNA. **(D, E, F)** Structure I is shown in (D), structure II is shown in (E), and structure III is shown in (F). **(G, H, I)** Zoomed-out views showing that the presence of Val-tRF bound to the 30S ribosomal SSU can cause steric hindrance of the ribosomal large subunit and therefore inhibit 70S association. **(G, H, I)** structure I; (H) structure II; (I) structure III.

process is assisted by aIF1A (58). Docking aIF1A from a previously determined archaeal translation initiation complex onto the current Val-tRF structure model indicates that the Val-tRF D loop in structures I and II partially overlapped with the binding site of aIF1A, which is located at the A-site of 30S (54) (Fig S8). This suggests that aIF1A and Val-tRF's binding on the 30S ribosome are mutually exclusive. Moreover, we found that in all three complex structures, Val-tRFs clashed with the mRNA P-site codon (Fig 5A–C). The binding interactions of the mRNA in the A, P, and E sites of the 30S ribosome are relatively conserved, whereas the binding of upstream and downstream sequences is relatively flexible (34, 37, 38, 41, 43). Examination of these structures thus indicated that Val-tRNA can interfere with the placement of the AUG codon at the P-site during the initiation of translation.

## Interference with 70S ribosome assembly

Docking of the 70S *Sac* ribosome structure onto structures I, II, and III revealed a steric hindrance between Val-tRF and H69 of 23S rRNA (Fig 5D–F). This incompatibility with H69 indicates that Val-tRF may influence the assembly of the 30S SSU with the 50S large subunit to form 70S ribosomes (Fig 5G–I). In this regard, the findings of a previous study have indicated that Val-tRF inhibits 70S ribosome translation by interfering with peptidyl transferase activity (32), thereby indicating that Val-tRF can also bind to the 70S ribosome. As shown in each of the three complex structures, Val-tRF adopts a distinct conformation that facilitates binding to different positions on the 30S SSU. It is thus plausible that Val-tRF is also characterized by differing patterns of binding to the

70S ribosome, in which Val-tRF no longer clashes with the H69 of 23S rRNA. Previous studies have, nevertheless, shown that Val-tRFs bind primarily to the 30 SSU (32, 33), and only under conditions of high pH stress can Val-tRF bind to polysomes in cell lysates. The findings of our structural analysis would tend to indicate that Val-tRF binding to the SSU can interfere with the formation of the 30S initiation complex and 70S assembly, which is consistent with the findings of a functional study that has provided evidence to indicate that tRFs can contribute to the regulation of gene expression by fine-tuning the rate of translation (33).

# Discussion

The findings of our structural analysis of a Val-tRF-derived fragment are consistent with previous functional characterizations of Val-tRFs (32, 33). The binding of Val-tRF displaces mRNA from the translation initiation complex, resulting in a global initiation of translation both in vivo and in vitro. Upon binding to Val-tRF, we observed a steric hindrance of P-site mRNA codons, incompatibility with aIF1A, and a disordered anti-SD sequence in 16S rRNA. Collectively, these effects contribute to disrupting the initiation process during translation. The fact that archaeal Val-tRFs also inhibit eukaryotic and bacterial protein translation tends to indicate a functionally conserved mode of action (33). Our structural observations provided evidence to indicate certain conserved interactions. Notably, the binding sites of Val-tRF, including the decoding sites, h28 and h24, appear to be highly conserved across the three domains of life. Moreover, there appears to be a conservation of the residues in the ribosomal SSU, with which Val-tRF interacts. For example, in structure I, structural modeling and refinement indicated that the G496 residue (a highly conserved base in decoding center) of 16S rRNA comes into contact with the G20 residue of Val-tRNA.

In our observations from cryo-EM 3D classifications, we captured the multiple distinct conformations of Val-tRF. We first ruled out that the conformational variation was caused by inactive 30S ribosomes. We demonstrated that the 30S is active through an in vitro poly-phenylalanine synthesis assay (Fig S9). Variation of Val-tRF conformation might be associated with the lack of in vivo RNA modifications in in vitro transcribed tRNA. Certain modifications in tRFs can enhance the stability of RNA duplexes by strengthening base-stacking interactions. Modifications may also alter the accessibility of binding sites on the tRF surface, thereby influencing the interactions of related proteins or ribosomal rRNA. Nevertheless, as we found in structures I, II, and III, the interacting components of the ribosome with Val-tRF are very conserved, while the overall conformation of Val-tRF varies. In addition, Val-tRF in all structures destabilizes several helices in close proximity to the decoding site (h24, h44, h45). These also suggest that the mechanism we discovered for in vitro-transcribed Val-tRF inhibiting protein translation is similar to the mechanism of action of in vivo Val-tRF.

# Materials and Methods

### Ribosome purification from *S. acidocaldarius* DSM639

Ribosomes were prepared from the archaeon *S. acidocaldarius* (*Sac*) DSM639, as follows. Cells were initially grown at 75°C and pH 3.5. To obtain *Sac* ribosomes, 2*g* samples of cell pellets were harvested and resuspended in 30 ml of buffer A (25 mM Tris–HCl pH 7.5, 100 mM $NH_4Cl$, and 10.5 mM $MgCl_2$) at 4°C, lysed twice (18 kPa; ConstantSystem), and centrifuged at 16,300*g* (HITACHI R20A2) for 40 min. The supernatants thus obtained were loaded onto a 37.7% sucrose cushion (20 mM Tris–HCl pH 7.5, 37.7% sucrose, 100 mM $NH_4Cl$, and 10.5 mM $MgCl_2$) and centrifuged in a Ti45 rotor (Beckman) at 167,900*g* for 21 h at 4°C. Subsequently, the pellets were suspended in buffer A, loaded onto a 10–35% sucrose gradient, and spun in an SW32 rotor (Beckman) at 68,300*g* for 13 h at 4°C. Fractions containing the 30S subunits were collected using a Biocomp Piston Gradient Fractionator and pelleted separately after centrifugation in a Ti45 rotor (Beckman) at 167,900*g* for 17 h at 4°C. Finally, the pellets were suspended in buffer B (25 mM HEPES KOH pH 7.5, 100 mM $NH_4Cl$, 10 mM $MgCl_2$, and 1 mM DTT), flash-frozen in liquid nitrogen, and stored at –80°C until further use.

### Minimal polyphenylalanine synthesis assay

In order to test the activity of 30S subunit, one direct method is to use in vitro translation assays where the 30S subunit is combined with the necessary components for translation, including mRNA, tRNAs, amino acids, the 50S ribosomal subunit, and various translation factors. The production of a polypeptide product indicates that the 30S subunit is active in vitro. tRNAPhe was charged by adding 2,000 pmol tRNAPhe, 10 nmol [14C]-Phenylalanine (509.0 mCi/mmol, total 50 µCi 500 µl; PerkinElmer), 4 mM ATP, and S100 enzymes (25 mg/ml) in a buffer containing 50 mM Hepes K (pH 7.5), 10 mM $MgCl_2$, 50 mM KCl, and 5 mM DTT; the reaction was performed in a total volume of 50 µl at 37°C for 30 min. The $^{14}C$-Phe-tRNA$^{Phe}$ was extracted with phenol (pH 5.3)-chloroform, followed by ethanol precipitation; then, $^{14}C$-Phe-tRNA$^{Phe}$ was resuspended in 2 mM sodium acetate (pH 5.3) for further use. For Sac poly-Phe synthesis assay, 40 pmol 30 s, 40 pmol 50 s, 1,000 pmol poly(U), $^{14}C$-Phe-tRNA$^{Phe}$ (200 pmol), 400 pmol EF-G, 400 pmol EF-tu, 400 pmol EF-Ts, and 4,000 pmol GTP were incubated in a buffer containing 20 mM Hepes K (pH 7.5), 18 mM $MgCl_2$, 10 mM NH4Cl, and 3 mM spermine at 70°C for 10 min in a total volume of 10 µl. We incubated the complex for 20, 30, and 40 min, respectively, for comparing the effect of different time scales on poly-Phe synthesis. Finally, a total of four samples (10 µl/each) were spotted on Whatman GF/C (GE) filter papers, which were incubated on ice for 20 min with 10% TCA (2 ml per filter) and washed three times with 5% TCA at RT for 5 min followed by washing twice with 95% ice ethanol at RT for 5 min. The dried filters were placed in measuring containers with 10 ml of scintillation fluid; after 12 h shielded from the light, the radioactivity was measured on a liquid scintillation meter for 5 min. For data processing, the $^{14}C$ content in the sample was converted as 1 DPMI = 1/2, 200,000 µCi; then, the value of $^{14}C$ incorporation fraction was obtained by the amount of obtained $^{14}C$-labeled Phe/total tRNA

added. The polyphenylalanine fraction incorporated into TCA-precipitable poly-phe over time (in min) indicates.

### Val-tRF preparation

The synthesis of Val-tRF used a specific double-stranded DNA sequence, 5′-GAAATTAATACGACTCACTATAGGAAAAGGGUUGGUGGUCUA-GUCUGGUUAUGA-3′, which was synthesized by Integrated DNA Technologies (with the annealing of commercially sourced complementary oligonucleotides to construct a double-stranded DNA). This sequence includes a T7 promoter for T7 RNA polymerase binding during transcription and incorporates a specific 26 nucleotide (nt) Val-tRF sequence initiated by GGGUU. A 500 $\mu$l in vitro transcription reaction was set up containing 50 $\mu$l annealing product, 45 $\mu$L 10x transcription buffer (250 mM Tris–Cl, pH 7.5; 250 mM Tris–Cl, pH 8.8; 240 mM $MgCl_2$; 10 mM spermidine; 0.1% Triton X-100; 50 mM DTT), 80 $\mu$l NTP mix (100 mM), 25 $\mu$l RNA polymerase inhibitor, 3 $\mu$l T7 RNA polymerase (21 mg/ml), and 297 $\mu$l nuclease-free water. The reaction was incubated at 37°C for 5 h. The transcriptional output from this sequence underwent purification via preparative urea PAGE, using a gel composition of 9.5% polyacrylamide with 1× TBE buffer, supplemented with 7 M urea. Following electrophoresis, the Val-tRF fraction was isolated through phenol–chloroform extraction and precipitated with 100% ethanol. The purified RNA samples were then reconstituted in RNase-free water, subjected to rapid freezing, and stored at -80°C for further investigative pursuits.

### Formation of a 30S-Val-tRF complex

For the purposes of generating a 30S–Val-tRF interaction complex, the Val-tRF was initially refolded by heating for 30 s at 65°C and then cooled for 10 min at RT. Thereafter, *Sac* 30S ribosomes (80 pmol) were incubated with 10-fold Val-tRF (800 pmol) at 70°C for 20 min in a 20-$\mu$l reaction system containing 20 mM HEPES KOH (pH 7.5), 150 mM $NH_4Cl$, 6 mM $Mg(OAc)_2$, 2 mM spermidine, 0.05 mM spermine, and 4 mM $\beta$-ME. Finally, the complexes were immediately transferred to ice and diluted for cryoelectron microscopy.

### Cryo-EM data acquisition

The 30S–Val-tRF complex generated as described in the previous section was incubated with 2% trehalose to protect the samples from damage during freezing and to enhance the resolution of cryo-EM images. Quantifoil holey carbon grids (R1.2/1.3, 300 mesh, Cu+2 nmC) were glow-discharged at 15 mA for 25 s, and a vitrobot Mark IV semi-automated plunge-freeze instrument (FEI) was then pre-equilibrated at 16°C and 100% humidity for 20 min before plunging. Aliquots (3 $\mu$l) of ~0.2 mg/ml samples were applied to individual prepared grids and subsequently flash frozen in liquid nitrogen-cooled liquid ethane with blotting for 4 s.

To confirm the formation of a 30S–Val-tRF complex, we initially inspected our grids and obtained a few micrographs for preliminary structural analysis using a 200 kV FEI TECNAI microscope. The density of Val-tRF was clearly observed in the low-resolution structure depicted in the micrographs. Subsequently, the grids were imaged using a Titan Krios electron microscope operated at 300 keV and equipped with a Gatan K3 Summit detector and GIF

Quantum energy filter. Movie stacks were automatically collected using AutoEMation software (written by Jianlin Lei), and images were recorded at ×81,000 magnification with a defocus range of −1.5 to −2.0 $\mu$m. Thereafter, each stack was exposed for 2.56 s, for an exposure time of 0.08 s per frame, resulting in 32 frames per stack. The total dose rate for each stack was estimated to be ~50 e⁻/Å2, and the pixel size was 1.087 Å/pixel.

### Image processing

The cryo-EM images were briefly processed using the CryoSPAC program suite. To deal with the data processing, 6,035 movie files were processed using patch motion correction, there yielding .mrc files. For the CTF parameters, "Patch CTF estimation" was used to calculate the values. For particle picking, we used crYOLO. For the picking model, we used cryo-EM images, which had been trained on low-pass-filtered images. The output .star files were combined into a single .star file, which was loaded into CryoSPAC. The "Extract From Micrographs" program of this software facilitates the combination of CTF values and the coordinate information from the .star files for particle extraction. A total of 854,284 particles were thus extracted from the micrographs. Following 2D classification (setting the number of 2D classes as 200), the 30S SSU was selected and used for "Ab-Initio reconstruction" and "homogeneous refinement." The refined map indicated a well-solved 30S density with a clear secondary structure and a visible portion of the 16S rRNA base pairs. However, the 30S head density was fuzzy. This map also enabled us to preliminarily identify the binding site of Val-tRF on the 30S SSU. Thus, the aligned particles from "homogeneous refinement" were subjected to particle sorting using a focus mask on the 30S head or Val-tRF. Three-dimensional classification of the 30S head identified a major conformation comprising ~60.1% of the particles. With a fixed head orientation, the focus mask of Val-tRF was used to sort the different conformations and slightly different ligand-binding positions. Three major classes (I, II, and III) were selected for refinement, which yielded three reconstructions comprising 9,742, 5,281, and 6,795 particles, respectively. For the purposes of refinement, each structure was subjected to homogeneous, local CTF, and global CTF refinements.

### Model construction

The *Sac* 30S model (PDB: 8HKX), which had been refined again into a 3.1 Å map, was initially docked into a refined map of the structure. Given differences in the orientation of the 30S head, the head and body domains of the 30S SSU were docked separately using "Fit in map" in Chimera. Furthermore, given the absence of the density of helices h24, h44, and h45, the 16S rRNA regions that were lacking in density were removed from the 30S model. The *E. coli* Val-tRNA model (residues 1–26, PDB: 4W29) was used as the starting model for Val-tRF, having initially mutated certain residues to be consistent with *H. volcanii* Val-tRF sequence. The initial model was docked onto the maps of the three structures. Manual refinement using Coot software was applied to modify the different conformations of the ligand in the three structures. The ligand and 30S models were combined and subjected to a single round of simulated annealing

refinement to rectify local structural changes. Secondary structure and base-pair restraints were incorporated via the refinement process. The output models obtained following annealing refinement were loaded into Coot to perform local structural adjustments. Subsequently, the models were used as input models for real-space refinement in Phenix by applying the minimization, local grid search, and adp refinement strategies. This process yielded the final refined models of structures I, II, and III.

## Data Availability

The cryo-EM maps for structures I, II, and III have been deposited in the EMDB with accession numbers EMD-37733, EMD-37734, and EMD-37604, respectively. The atomic models of these structures have been deposited in the PDB with accession numbers 8WQ2 (structure I), 8WQ4 (structure II), and 8WKP (structure III).

## Supplementary Information

## Acknowledgements

We thank Dr. Xiang-Wei He for the valuable discussions and the staff at the Center for Cryogenic Electron Microscopy, Zhejiang University, China. This research was supported by grants from the Natural Science Foundation of China (31971226), the Zhejiang Natural Science Foundation (LR20C050003), the Fundamental Research Funds for the Central Universities (2018QN81010), new faculty start-up funds from Zhejiang University, and the Thousand Young Talents Plan of China.

### Author Contributions

Y Wu: resources, data curation, software, formal analysis, validation, visualization, and methodology.
M-T Ni: data curation and methodology.
Y-H Wang: resources, data curation, software, formal analysis, and methodology.
C Wang: data curation and methodology.
H Hou: data curation and methodology.
X Zhang: data curation, software, formal analysis, and supervision.
J Zhou: conceptualization, data curation, supervision, funding acquisition, validation, investigation, project administration, and writing—original draft, review, and editing.

### Conflict of Interest Statement

The authors declare that they have no conflict of interest.

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
