## [Reviewer comments · Life Science Alliance]

Life Science Alliance

Structural basis of translation inhibition by a valine tRNA-derived fragment

Yun Wu, Meng-Ting Ni, Ying-Hui Wang, Chen Wang, Hai Hou, Xing Zhang, and Jie Zhou

DOI: <https://doi.org/10.26508/lsa.202302488>

Corresponding author(s): Jie Zhou, Zhejiang University and Xing Zhang, Zhejiang University

Review Timeline:

Submission Date:	2023-11-18
Editorial Decision:	2023-12-27
Revision Received:	2024-02-27
Editorial Decision:	2024-03-11
Revision Received:	2024-03-22
Accepted:	2024-03-22

Transaction Report:

December 27, 2023

Re: Life Science Alliance manuscript #LSA-2023-02488-T

Dr. Jie Zhou
Zhejiang University
Life Sciences Institute
866 Yuhangtang Rd.
Hangzhou, Zhejiang 310058
China

Dear Dr. Zhou,

Thank you for submitting your manuscript entitled "Structural basis of translation inhibition by a valine tRNA-derived fragment" to Life Science Alliance. The manuscript was assessed by expert reviewers, whose comments are appended to this letter. We invite you to submit a revised manuscript addressing the Reviewer comments.

Thank you for this interesting contribution to Life Science Alliance. We are looking forward to receiving your revised manuscript.

Sincerely,

B. MANUSCRIPT ORGANIZATION AND FORMATTING:

Reviewer #1 (Comments to the Authors (Required)):

The manuscript submitted by Wu et al. present structural analyses of a particular tRNA-derived RNA fragment originating from the 5' end of *H. volcanii* Val-tRNA in complex with the 30S ribosomal subunit from *Sulfolobus acidocaldarius*. This Val-tRNA-derived 26 residue long fragment (called Val-tRF) has previously been identified and functionally characterized by others. It has been suggested to inhibit translation, reduce peptide bond formation and compete with mRNA binding to archaeal ribosomes. The new cryo-EM structures presented herein provide unprecedented insight into the mode of action of this tRNA fragment. The authors show three different Val-tRF/30S subunit complexes that, however, all have in common that Val-tRF binds close to universally conserved 16S rRNA residues in the decoding center and furthermore destabilizes several helices in close proximity of the decoding site (h24, h44, h45).

This is the first structure of a ribosome-targeted tRF and thus this report clearly advances the field of regulatory ncRNA research. In particular the authors could deepen our understanding of the mode of action of Val-tRF on the archaeal ribosome and provide structural explanations for the previously reported functional data on Val-tRF.

In order to further improve this submission, the authors need to address the following points:

- 1) Very recently the field of tRNA-derived fragments has witnessed a landmark paper on a unifying nomenclature of tRFs (Holmes et al., Nature Methods, 2023, Vol. 20; <https://doi.org/10.1038/s41592-023-01813-2>) It is thus important and necessary that the authors of this submission follow this novel nomenclature and they should (at least once in their manuscript) use the standardized naming tDR (for tRNA-derived RNA).
- 2) Abstract, lines 27-28: it is unclear if the authors mean Val-tRNA or Val-tRF in this sentence. I think the authors mean the latter. This needs to be clarified.
- 3) Fig. 2G and H legend: it states "Val-tRNA" but I am certain that the authors actually meant "Val-tRF" instead. This needs to be changed.
- 4) Lines 247-251 & Suppl. Fig. 8: The authors should make clear in the text and the figure, that the structure of aIF1A was derived from a different publication (I assume) and docked onto their 30S cryo-EM densities. It is also necessary to give explicit reference to the relevant aIF1A publication.

Minor points:

- 1) Lanes 35-37: it is kind of odd that the authors refrain from listing ribozymes when they summarize the most important functions of non-coding RNA
- 2) Line 323: I assume it should read "HEPES KOH" and not "HEPES K"

Reviewer #2 (Comments to the Authors (Required)):

The manuscript „Structural basis of translation inhibition by valine tRNA-derived fragment" provides a structural explanation for the inhibitory function of the Val-tRF on protein synthesis described in previous papers. The results described offer interesting potential explanations for the functional role of tRFs during gene expression. Using cryo-electron microscopy, the authors generated structures of the small ribosomal subunit of *Sulfolobus acidocaldarius* in combination with Val-tRF and observed the formation of 3 major complexes with Val-tRF in different conformations. The proposed binding site of the tRNA fragment and its impact on the ribosomal structure suggest why Val-tRF is inhibitory for protein synthesis. Val-tRF is located at conserved RNA binding sites and thus blocks tRNA binding, aIF1A binding and destabilizes helices involved in the decoding of the mRNA/tRNA interactions. In addition, the 70S assembly is potentially disturbed in the presence of the tRNA fragment.

Overall, the manuscript provides a valuable insight into the function of tRNA fragments and provide a mechanism of function. However, several points should be addressed to improve the clarity of the manuscript.

1. Where does the Val-tRNA sequence come from (*Haloferax* or *Sulfolobus*)? If they are different, can this affect the binding of the tRF? What about the differences between the *Haloferax* and *Sulfolobus* ribosomes-are the binding sites identical?
2. The authors state that only two base pairs (line 101) are formed between the ribosome and the tRF. Which one are those?

Are these really sufficient to stabilize the tRF on the ribosome? Maybe the wording is misleading here (or I misunderstood). Are these interactions maintained in the three reported variations of the tRF?

3. What about modifications of the tRFs. I am aware that not much is known about them in tRFs, but if only a few interactions are formed, it should be discussed if and how they could affect the overall function/structure.
4. Fig. 2F: Where does this structure come from? Is the extracted tRF from one of the reported structures? Please clarify what is shown.
5. Line 132/136: It is not clear to me what the authors would like to state. Please rephrase/clarify.
6. Line 139/141: I am not sure if the limitation in the number of building blocks of RNA really limits the possible interactions of RNA in comparison to proteins. RNAs and RNA aptamers can also be highly specific. I guess it depends on point of view.
7. The authors state that the tRF mimics the tRNA anticodon stem loop. I am not sure if this is the correct phrasing. Although it is the same binding site, the structures are somewhat different and to my knowledge not really mimic a tRNA/mRNA anticodon.
8. Line 315: I guess a typo: not an mRNA was extracted but the tRF.
9. The Val-tRF preparation needs to be more detailed. The provided DNA sequence does not contain a T7 Promotor sequence. In addition, a completely single stranded DNA sequence is not recognized by the RNA polymerase. Was the DNA amplified, part of a cloned vector or how was the T7 transcription performed. Please provide more details. Were methylated reverse primer used to avoid the non-templated addition of As to the transcript?
10. Is there any functional evidence that the 30S subunit is functional after the purification? Is there a possibility for quality control?
11. The authors state that the tRF destabilizes h44, h45 and h24. However, they show the helices in their structure with the tRF (4B). In 4C the steric clash between the tRF and the helices is depicted. How is the overall structure changed in respect to h24, h45, h44 when comparing the subunits with and without tRFs? According to 4C it overlaps considerably and where do these residues go and how are they rearranged to fit the tRF? Is there any information on the binding affinity? It is impressive that the tRF can rearrange rather large proportions of the helices.

Reviewer #3 (Comments to the Authors (Required)):

In the manuscript entitled « Structural basis of translation inhibition by a valine tRNA-derived fragment", Wu et al present three different cryo-EM structures of Val-tRF non-coding RNA bound to an archaeal small ribosomal subunit. Overall, the results allow for the reconstruction of robust models. However, they only represent 2.5% of the total amount of particles, questioning their biological significance.

Major comments

Abstract

The authors claim that archaeal Val-tRF is active in all domains of life. This is not clear to me: do tRFs exist in all domains of life or is it simply a result obtained by in vitro approaches? In that case I do not see the importance of such a statement.

Introduction

Page 2, l59. Val-tRFs interact with 30S but also 70S ribosomes. Though, all the following experiments study their interactions with isolated small subunits only (see below, remark on p4). Please comment this choice.

Results and Discussion

Page 2, l84: Val-tRF are in vitro transcribed. Are their "real" counterparts modified in vivo? If yes would these modifications influence their binding capabilities?

Page 3: the authors identified 30 different Val-tRF conformations, reflecting the high flexibility of the RNA. However, they decided to choose and solve only three structures over the 30, based on the (very) relative higher number of particles. The number of particles is still very low (<10000) and the global resolutions rather poor regarding the current standards.

Do this reflect any biological significance of the three selected models over the other ones?

Page 4, top. Could the interactions and differences between the various models described be due to the absence of the large subunit? Are the structures described on the 30S SSU compatible with 70S ribosomes? Indeed, Page 7, l272 the authors state that Val-tRF can interfere with 70S assembly while in the introduction the state that it binds to 70S.

Page 4, l150: show the similarities with ASL binding on a figure

Page 5, l208 and so on. The description of 30S modifications induced by the binding of Val-tRF suffer from the absence of a real

"wet" control, using the same ribosomes and cryo-EM procedure. In other words, a map without Val-tRF is necessary here to check whether the h24, h44 and h45 helices are not altered. It could simply come from empty 30S particles that were not used in the final reconstructions.

Overall global resolutions are rather poor, which can be explained by the small amounts of particles and the dynamics of the 30S. However, what are the local resolutions of the Val-tRF described in the three complexes? If they are too low did the authors use rigid body fitting? What is the precision of their atomic models?

Methods

Do the authors use motioncor2 (line 344) and then patch motioncorrection in cryosparc (line 350)? Please explain

Line 349 : Tilted data ? Please explain: did the authors used tilted series?

I am a bit confused by the particle numbers:

Starting from 854284 particles the authors keep only $9742 + 5281 + 6795 = 21818$ particles to reconstruct the three major classes. That means that 97.5% are not used? How can the authors claim that only 2.5% of the particles are biologically relevant, without overinterpretation?

Reviewer #1 (Comments to the Authors (Required)):

The manuscript submitted by Wu et al. present structural analyses of a particular tRNA-derived RNA fragment originating from the 5' end of *H. volcanii* Val-tRNA in complex with the 30S ribosomal subunit from *Sulfolobus acidocaldarius*. This Val-tRNA-derived 26 residue long fragment (called Val-tRF) has previously been identified and functionally characterized by others. It has been suggested to inhibit translation, reduce peptide bond formation and compete with mRNA binding to archaeal ribosomes. The new cryo-EM structures presented herein provide unprecedented insight into the mode of action of this tRNA fragment. The authors show three different Val-tRF/30S subunit complexes that, however, all have in common that Val-tRF binds close to universally conserved 16S rRNA residues in the decoding center and furthermore destabilizes several helices in close proximity of the decoding site (h24, h44, h45). This is the first structure of a ribosome-targeted tRF and thus this report clearly advances the field of regulatory ncRNA research. In particular the authors could deepen our understanding of the mode of action of Val-tRF on the archaeal ribosome and provide structural explanations for the previously reported functional data on Val-tRF.

Response: We express our gratitude to the reviewers for acknowledging the importance of our structural investigations. Our goal is to decipher the structural foundation that underpins the outcomes of prior functional experiments through structural analysis. Our structures enable a deep understanding of molecular mechanisms, unveiling the manner in which Val-tRF competes with mRNA for binding and how Val-tRF obstructs the initiation of translation.

In order to further improve this submission, the authors need to address the following points:

1) Very recently the field of tRNA-derived fragments has witnessed a landmark paper on a unifying nomenclature of tRFs (Holmes et al., Nature Methods, 2023, Vol. 20; <https://doi.org/10.1038/s41592-023-01813-2>)

It is thus important and necessary that the authors of this submission follow this novel nomenclature and they should (at least once in their manuscript) use the standardized naming tDR (for tRNA-derived RNA).

Response: We are grateful to the reviewers for reminding us to use the most up-to-date nomenclature for tRNA derived fragment. Accordingly, we used tDF-1:26-Val-GAC-1 for Val-tRF and have clarified this in the manuscript introduction section.

2) Abstract, lines 27-28: it is unclear if the authors mean Val-tRNA or Val-tRF in this sentence. I think the authors mean the latter. This needs to be clarified.

Response: We apologize for this typo. It should be Val-tRF in the sentence.

3) Fig. 2G and H legend: it states "Val-tRNA" but I am certain that the authors actually meant "Val-tRF" instead. This needs to be changed.

Response: We thank the reviewer for pointing out the typo. Yes, it should be Val-tRF instead of Val-tRNA.

4) Lines 247-251 & Suppl. Fig. 8: The authors should make clear in the text and the figure, that the structure of aIF1A was derived from a different publication (I assume) and docked onto their 30S cryo-EM densities. It is also necessary to give explicit reference to the relevant aIF1A publication.

Response: We appreciate the reviewer's feedback on the unclear description of aIF1A. The manuscript and figure legends have been updated accordingly. We utilized the aIF1A model from a previous archaeal translation complex study (Coureux et al., Nature Communications, 2016) as a docking model to accurately position aIF1A in the current Val-tRF-30S complex model. Additionally, we have now cited the aIF1A publication in the text to provide a clearer reference for our readers.

Minor points:

1) Lanes 35-37: it is kind of odd that the authors refrain from listing ribozymes when they summarize the most important functions of non-coding RNA.

Response: We appreciate the reviewer's reminder regarding the ribozyme reference. It has now been included in the manuscript.

2) Line 323: I assume it should read "HEPES KOH" and not "HEPES K"

Response: Yes, it should be "HEPES KOH", We have corrected it.

Reviewer #2 (Comments to the Authors (Required)):

The manuscript „Structural basis of translation inhibition by valine tRNA-derived fragment" provides a structural explanation for the inhibitory function of the Val-tRF on protein synthesis described in previous papers. The results described offer interesting potential explanations for the functional role of tRFs during gene expression. Using cryo-electron microscopy, the authors generated structures of the small ribosomal subunit of *Sulfolobus acidocaldarius* in combination with Val-tRF and observed the formation of 3 major complexes with Val-tRF in different conformations. The proposed binding site of the tRNA fragment and its impact on the ribosomal structure suggest why Val-tRF is inhibitory for protein synthesis. Val-tRF is located at conserved RNA binding sites and thus blocks tRNA binding, aIF1A binding and destabilizes helices involved in the decoding of the mRNA/tRNA interactions. In addition, the 70S assembly is potentially disturbed in the presence of the tRNA fragment.

Overall, the manuscript provides a valuable insight into the function of tRNA fragments and provide a mechanism of function. However, several points should be addressed to improve the clarity of the manuscript.

Response: We are grateful for the reviewer's positive feedback on the significance of the three

structures We clarified the role of Val-tRF in gene expression by solving the structures of Val-tRF bound with 30S ribosome subunit. Our results are consistent with previous functional studies.

1. Where does the Val-tRNA sequence come from (Haloferax or Sulfolobus)? If they are different, can this affect the binding of the tRF? What about the differences between the Haloferax and Sulfolobus ribosomes-are the binding sites identical?

Response: The Val-tRF sequence originates from *Haloferax volcanii*, while the 30S subunit is derived from *Sulfolobus acidocaldarius* (Sac). We opted for the Sac 30S ribosome due to its relative stability and suitability for cryo-EM studies. Val-tRF binds to highly conserved components on the 30S ribosome, including the decoding center, h23, and the 30S neck helix. As depicted in Figure 1, residues such as G496 (in the decoding center), G659, A660 (in helix 23), and G895, G896 (in helix 28 on the 30S neck) are highly conserved and engage in direct interactions with Val-tRF. The binding sites of Val-tRF are consistent across *Haloferax* and *Sulfolobus*.

CLUSTAL O(1.2.4) multiple sequence alignment

```
Sulfolobus      GCGCCCGAUUCCGGUUAUCUCCGCGCCGACCGCUAUCGGGGUAGGUAAGCCAU 60
Haloferax      GCGACCAUUCGGUUAUCUCCGCGGAGGUAUUGCUAUGGGGUCGUAUAGCCAU 60
                ** ***** * * * * *
Sulfolobus      GGAGUCUACACUCCGGUUAAGGAGUUGGCGGACGGUCGUAACACGUGGCUAAC 120
Haloferax      CUADUUGACG-----AGUUCUACUCUGGCGAAAGCUCAGUAACCGUGGCCAAC 120
                *** ** * * * * *
Sulfolobus      UACCCUUCGGGACGGGUAUACCCGGGAAACUGGGUAUUAUCCCGAUAGGGAAGGAGUC 180
Haloferax      UACCCUACAGAGAACGUAUACCCGGAACUGAGGCUUAUUAUACGCGGAGUACUG 174
                ***** ** ***** * * * * *
Sulfolobus      CUGGAUUGGUUCUUCUUAAGGCUUAGGUUUCUCCGUUUGAGCCGCCGAGGA 240
Haloferax      CUGGAUUGGCUUUCUUAAGGCUUAGGUUUCUCCGUUUGAGCCGCCGAGGA 214
                ***** ** * * * * *
Sulfolobus      UGGGCUACGGCCAUACGCGUGGCGGUAAGGCGCACGAAACCUUAUACGGGUA 300
Haloferax      UGUGGCUUGCGCCGUAUAGGUAGAGCGGUGGUAACGCCACCGGCGGUAUUCGUA 274
                ** * * * * *
Sulfolobus      GGGCCUGGAGAGCGAGCCUCCAGUUGGGGACUAGACAGGGCCAGGCGUACGG 360
Haloferax      CGGUUUGAGAGCAAGACCCGGAGCGAAUUCGAGCAAGUUCGGCCUACGG 334
                *** ** * * * * *
Sulfolobus      GCGCACAGGCGGAAACGUCUUAAGGCGGAGGCGUAGGCGUACCCGAGUGCC 420
Haloferax      GCGCACAGGCGGAAACCUUUAACUACAGCAGGUAAGGCGGAAAGGCGGAAAGG 394
                ***** * * * * *
Sulfolobus      UCCGCAAGGAG----GCUUUCUCCGCUUUAAGGCGGGGUAUAGCGGGGCAA 475
Haloferax      AGGGCUAUAUAGUUCUUCGCUUUCGACGUAAGGCGGCGAGGUAUAGGCGGCAA 454
                ** * * * * *
Sulfolobus      CUCUGGUGUACCGCCGCGUAUUAACACUCUCCGAGUGGCGGGUUAUUCGGCG 535
Haloferax      GACCGUGGACCGCCGCGUAUUAACCGGACUAGUAGUAGUAGUAGUAGUAGUAG 514
                * * * * *
Sulfolobus      CUAAGCGCUGUAGCGGCCCAAGUCGCCCUUAUAGUCCCGGCUACCGGGGA 595
Haloferax      CUAAGCGCUGUAGCGGCCCAAGUUAUAGUAGUAGUAGUAGUAGUAGUAGUAG 574
                ***** * * * * *
Sulfolobus      ACUUGGGGCG-AUACUUGGUGGCUAGGGGCGGAGAGCGGGGUAUCUCCGAGUAG 654
Haloferax      GCGUCGUGGUAUAAACCGUUCUUGGCGGAGGCGGAGGCGGCGGUAUCUCCGAGUAG 634
                * * * * *
Sulfolobus      GGGGAAAUUUAUUAUUAUUAUUAUUAUUAUUAUUAUUAUUAUUAUUAUUAUUA 714
Haloferax      GAGGAAAUUUAUUAUUAUUAUUAUUAUUAUUAUUAUUAUUAUUAUUAUUAUUA 694
                * * * * *
Sulfolobus      CCGGACGGUAGAGGCGAAAGCGGGGAGCAACGGGUAUUAUUAUUAUUAUUAUUA 774
Haloferax      UCCGACGGUAGAGGCGAAAGCGGGGAGCAACGGGUAUUAUUAUUAUUAUUAUUA 754
                ***** * * * * *
Sulfolobus      GCUUUAACGAUGCGGCUAGGUUCGAGUAGGCUUAGGCUUUCGUGGCGCCGAGGA 834
Haloferax      GCUUUAACGAUGCUUAGGUUCGAGUAGGCUUAGGCUUUCGUGGCGGCUUUAAGGA 814
                ***** * * * * *
Sulfolobus      AGCCGUUAAGCCCGCCGCGGAGGUAUAGGCGCAAGACUAAACUUAAGGAAUUGGC 894
Haloferax      AGCCGAGAAGCGAGCGCGGAGGUAUAGGCGCAAGACUAAACUUAAGGAAUUGGC 874
                * * * * *
Sulfolobus      GGGGAGACACACAAGGGGUAACUUGCGGCUUAUUAUUAUUAUUAUUAUUAUUA 954
Haloferax      GGGGAGACACUUAACCGGAGGCGGCUUUAUUAUUAUUAUUAUUAUUAUUAUUA 934
                * * * * *
Sulfolobus      CCGGGGAGACCGCAGU-AUGACGGCCGAGCUAACGCUUUGCGU-AC-UUGCGGAGAG 1011
Haloferax      CCAGUCGCUUACAGUAGGAGCUAGGCUUUAUUAUUAUUAUUAUUAUUAUUAUUA 994
                ** * * * * *
Sulfolobus      GAGGUGCAUGGCCCGCCGCGUUGUUGAAUUGCGUUAAGUCCGGCAACGAGC 1071
Haloferax      GAGGUGCAUGGCCCGCCGCGUUGUUGAAUUGCGUUAAGUCCGGCAACGAGC 1054
                ***** * * * * *
Sulfolobus      GAGACCCACCCUUAUUGUUAU----UCUGGACUCCGUCGAGAACACACUAGGG 1125
Haloferax      GAGACCCACCUUAUUGUUAUUGGAGCAGUUAUUGGAGU-----CUGGUAUUAAGA 1109
                ***** * * * * *
Sulfolobus      GGACUGCCGCG-GUAAGCCGAGGAGGAGGCGCACGCGAGGCUAGCAUAGCCCGAAA 1184
Haloferax      GGACUGCCGCUUAAGCGGAGGAGGAGGCGCACGCGAGGCUAGCAUAGCCCGAAA 1169
                ***** * * * * *
Sulfolobus      CUCCGGGCGCACCGGGUUAUUAUUGGAGGACACGGGUAUUAUUAUUAUUAUUA 1244
Haloferax      GAGCUGGGCUACACGGGCUUAUUAUUGGAGGACACGGGUAUUAUUAUUAUUAUUA 1229
                * * * * *
Sulfolobus      AGCCAUC-CUUAACCUCCGCGAGUUGGAGUAGGCGUAAACCCGCCUUGUAGAC 1303
Haloferax      CGCUAUCUUAACCUCCGCGAGUUGGAGUAGGCGUAAACCCGCCUUGUAGAG 1289
                ** * * * * *
Sulfolobus      GAGGAUUCUUAUUAACCGGGUUAACACCGCGGUAUUAUUAUUAUUAUUAUUA 1363
Haloferax      CUGGAUUCUUAUUAACCGGGUUAACACCGCGGUAUUAUUAUUAUUAUUAUUA 1349
                ** * * * * *
Sulfolobus      CACACCCCGCUGCUCACCGGAGGAAAGGGUAGGUCCUUGCGUAUAGGGGG 1423
Haloferax      CACACCCCGCUAAAGCACCGGAGGAGGCGGAGGCGCAC-----ACACGGU 1402
                ***** * * * * *
Sulfolobus      GAUCAACUUCUUCUCCGCGAG-GGGGAAAGUUAACAAGGUAAGCCUAGGGGAAACU 1482
Haloferax      GGUCGAUCUUGGCGUCCAGGGGGCUUAAGUUAACAAGGUAAGCCUAGGGGAAACU 1462
                * * * * *
Sulfolobus      GCGGCGUAGUACCUCAU 1500
Haloferax      GCGGCGUAGUACCUCCU 1480
                ***** * * * * *
```

2. The authors state that only two base pairs (line 101) are formed between the ribosome and the tRF. Which one are those? Are these really sufficient to stabilize the tRF on the ribosome? Maybe the wording is misleading here (or I misunderstood). Are these interactions maintained in the three reported variations of the tRF?

Response: We apologize for the misleading wording. It should be noted that the two base pairs (U11-A26, C12-G25) within the Val-tRF contribute to stabilizing the structure and conformation of the Val-tRF molecule. These two base pairs are preserved in the Val-tRF of all three structures. Accordingly, we have revised the text.

3. What about modifications of the tRFs. I am aware that not much is known about them in tRFs, but if only a few interactions are formed, it should be discussed if and how they could affect the overall function/structure.

Response: We thank the reviewer for suggesting the discussion of tRF modifications. Certain modifications in tRFs can enhance the stability of RNA duplexes by strengthening base stacking interactions. This increased stability can result in tighter binding between the modified tRF and its target RNA, leading to higher binding affinities. Modifications may also alter the accessibility of binding sites on the tRF surface, thereby influencing the interactions of r-proteins or rRNA. Changes in binding site accessibility can modulate the kinetics of complex formation and impact the overall binding specificity of the modified tRF. The discussion has been incorporated into the appropriate paragraph.

4. Fig. 2F: Where does this structure come from? Is the extracted tRF from one of the reported structures? Please clarify what is shown.

Response: The model present in Fig.2F comes from a previously determined crystal structure of Val-tRNA(PDB: 4W29). We have clarified this point in the figure legend.

5. Line 132/136: It is not clear to me what the authors would like to state. Please rephrase/clarify.

Response: We have revised the text from lines 132-136 to clarify our point. Our intention is to highlight that the differences in chemical properties between RNA and proteins result in the Val-tRF adopting multiple conformations.

6. Line 139/141: I am not sure if the limitation in the number of building blocks of RNA really limits the possible interactions of RNA in comparison to proteins. RNAs and RNA aptamers can also be highly specific. I guess it depends on point of view.

Response: We thank the reviewer to clarify that RNA molecule's interactions can also be highly specific. While it's true that RNA has fewer building blocks (four standard nucleotides) compared to proteins (which have twenty standard amino acids), this does not necessarily limit RNA's functional diversity or specificity of interactions. RNAs, including RNA aptamers, can fold into

complex three-dimensional structures that allow for highly specific interactions with other molecules, including proteins, small molecules, and other RNAs. This specificity is achieved through a combination of hydrogen bonding, stacking interactions, and shape complementarity, much like how proteins interact with their targets.

We have revised the wording in the text. Our intention is to convey that Val-tRF relies on just two base pairs to maintain its structure and folding, which results in multiple binding sites and conformations.

7. The authors state that the tRF mimics the tRNA anticodon stem loop. I am not sure if this is the correct phrasing. Although it is the same binding site, the structures are somewhat different and to my knowledge not really mimic a tRNA/mRNA anticodon.

Response: Yes, we agree that the structure of D stem-loop of Val-tRF is different from tRNA anticodon. We have deleted this statement in the text.

8. Line 315: I guess a typo: not an mRNA was extracted but the tRF.

Response: We thank the reviewer. Yes, it should be a typo. Val-tRF was extracted using phenol-chloroform.

9. The Val-tRF preparation needs to be more detailed. The provided DNA sequence does not contain a T7 Promotor sequence. In addition, an completely single stranded DNA sequence is not recognized by the RNA polymerase. Was the DNA amplified, part of a cloned vector or how was the T7 transcription performed. Please provide more details. Were methylated reverse primer used to avoid the non-templated addition of As to the transcript?

Response: In the methods section, we described the details of the Val-tRF transcription. Please refer to the methods section for the transcription procedure details.

10. Is there any functional evidence that the 30S subunit is functional after the purification? Is there a possibility for quality control?

Response: We appreciate the reviewer's emphasis on the importance of verifying the activity of the 30S ribosome. We have supplemented our work with experiments on in vitro translation activity using the 30S subunit.

11. The authors state that the tRF destabilizes h44, h45 and h24. However, they show the helices in their structure with the tRF (4B). In 4C the steric clash between the tRF and the helices is depicted. How is the overall structure changed in respect to h24, h45, h44 when comparing the subunits with and without tRFs? According to 4C it overlaps considerably and where do these residues go and how are they rearranged to fit the tRF? Is there any information on the binding affinity? It is impressive that the tRF can rearrange rather large proportions of the helices.

Response: We are grateful to the reviewer for highlighting the significant conformational changes in h24, h25, and h44. Upon comparing the 30S structure in the presence and absence of Val-tRF, we observed notable overall structural conformational differences (Figure 2A), especially in the head domain of 30S subunit. Specifically, for h44, adjacent components, such as uS5 and the G374 loop of the 16S rRNA, appeared to shift away from h44 (Figure 2B). In the case of h24, it was found to be distorted, contributing to its increased flexibility (Figure 2C). Regarding h45, the clash between the linker connecting h44 and h45 with the acceptor stem strand was observed, leading to increased flexibility in this region (Figure 2D). The destabilizing conformational effects on h24, h44, and h45 induced by Val-tRF binding are depicted in Figure 2D.

Figure 2 Conformational flexibility of 30S ribosome caused by Val-tRF binding. (A) cryo-EM density showing the overall conformational between Sac 30S ribosome in the presence (yellow) and in the absence of Val-tRF (green). The 30S head domain exhibits different conformations between these two structures. (B) Flexibility of h44 by binding Val-tRF induced displacement of uS5 and G374 loop. (C) Val-tRF induced distortion of h24. (D) Val-tRF caused unstable conformation of h24, h44 and h45, therefore no apparent density attribute to these components.

Reviewer #3 (Comments to the Authors (Required)):

In the manuscript entitled « Structural basis of translation inhibition by a valine tRNA-derived fragment », Wu et al present three different cryo-EM structures of Val-tRF non-coding RNA bound to an archaeal small ribosomal subunit. Overall, the results allow for the reconstruction of robust models. However, they only represent 2.5% of the total amount of particles, questioning their biological significance.

Response: We are thankful to the reviewer for considering the three structures we obtained as robust models. Through extensive 3D classification, we acquired over thirty maps of Val-tRF bound to the 30S subunit. Due to the limited number of particles available for cryo-EM reconstruction, only three of these structures achieved medium resolution, while the resolution of the remaining thirty-plus reconstructions was insufficient for accurate model building. However, as can be seen from the supplementary figure 4, the binding locations of Val-tRF in most of the additional thirty-plus structures are similar to those in structures I and II, positioned between the 30S G476 loop (G530 in E.coli) and h23, though the conformations of Val-tRF vary. We believe that structures I and II to some extent explain the mechanism of action observed in the other thirty-plus structures, namely the interaction of Val-tRF with the 30S subunit and the decoding center, inhibiting the binding of tRNA-mRNA and aIF1A to the ribosome, thereby suppressing protein translation.

Major comments

Abstract

The authors claim that archaeal Val-tRF is active in all domains of life. This is not clear to me: do tRFS exist in all domains of life or is it simply a result obtained by in vitro approaches? In that case I do not see the importance of such a statement.

Response: We agree with the reviewer that Val-tRF is unique to archaea and has the capability to inhibit protein translation in vitro in both eukaryotes and bacteria. We have revised the relevant statement in the abstract accordingly.

Introduction

Page 2, 159. Val-tRFs interact with 30S but also 70S ribosomes. Though, all the following experiments study their interactions with isolated small subunits only (see below, remark on p4). Please comment this choice.

Response: The three high-resolution structures of the Val-tRF-30S ribosome complex we analyzed, along with other low-resolution structures, all indicate that Val-tRF is incompatible with the 70S ribosome. This suggests that the binding sites and conformations of Val-tRF when bound to the 70S ribosome may differ from those when it is bound to the 30S subunit. Additionally, it is also possible that the limited number of conformations of Val-tRF bound to the 30S we captured did not include any that are compatible with the 70S ribosome.

Results and Discussion

Page 2, 184: Val-tRF are in vitro transcribed. Are their "real" counterparts modified in vivo? If yes would these modifications influence their binding capabilities?

Response: We appreciate the reviewer pointing out the differences between in vitro transcribed Val-tRF and its in vivo real modified counterparts. Currently, specific details on the modification residues of D stem loop Val-tRF in *Haloflex volcanii* are not available in the literature. These modifications could play roles in ensuring the stability and functionality of Val-tRF in high salt concentrations and temperatures, characteristic of the habitats of *Haloflex volcanii*. This might also partly explain why we were able to observe such a diverse range of conformations in the complex of in vitro transcribed Val-tRF with the 30S ribosome. This could be a limitation of our current study.

Page 3: the authors identified 30 different Val-tRF conformations, reflecting the high flexibility of the RNA. However, they decided to choose and solve only three structures over the 30, based on the (very) relative higher number of particles. The number of particles is still very low (<10000) and the global resolutions rather poor regarding the current standards.

Do this reflect any biological significance of the three selected models over the other ones?

Response: Using Val-tRF as a mask for 3D classification, we obtained structures of multiple distinct Val-tRF-30S complexes. As shown in the figure below, we selected nine structures that exhibited clear Val-tRF densities. The cryo-EM maps reveal that in these nine structures, Val-tRF is consistently located between the decoding center and h24. Furthermore, the binding of Val-tRF induces conformational flexibility in the 16S rRNA regions h24, h44, and h45. Based on the comparison of these structures, we consider Structures I and II to be representative of the Val-tRF-30S complex, with the main distinction from other structures being differences in the conformation of Val-tRF

Figure 3. Nine structures of Val-tRF bound with Sac 30S ribosome. In all these structures, h24, h44 and h45 are disordered. Also, Val-tRF locates between decoding center and h23.

Page 4, top. Could the interactions and differences between the various models described be due to the absence of the large subunit? Are the structures described on the 30S SSU compatible with 70S ribosomes? Indeed, Page 7, 1272 the authors state that Val-tRF can interfere with 70S assembly while in the introduction the state that it binds to 70S.

Response: A previous study has demonstrated that Val-tRF primarily binds to the small ribosomal subunit. Utilizing polysome gradient analyses and in vitro binding studies with *Haloflex volcanii* cell lysates, Val-tRF was shown to primarily co-migrate with the 30S subunit fraction (Gebetsberger, Archaea, 2012). Consequently, our structural studies have elucidated how Val-tRF binds to the 30S ribosome subunit to inhibit translation. As depicted in Figure 3, across 11 structures resolved by cryo-EM (9 structure in figure 3 and structure 1 and 2), Val-tRF is primarily situated between the decoding center and h23, although the conformation of Val-tRF varies. This variability in conformation can be attributed to two main factors. Firstly, there is a lack of sufficient internal base pairs to constrain its overall structure and conformation. Secondly, the in vitro transcribed Val-tRF may be missing modifications that are crucial for stabilizing its 3D structure. Considering that previous studies have indicated Val-tRF's ability to bind to the intact 70S ribosome, this suggests that the binding site on the 70S ribosome could be distinct from that on the 30S subunit.

Page 4, 1150: show the similarities with ASL binding on a figure

Response: The similarities between Val-tRF and ASL binding are illustrated in Figure 4 (see below). Furthermore, the comparable binding of ASL-mRNA and Val-tRF is shown in the main Figure 3A. page 16).

Figure 4 Comparison of tRNA ASL and Val-tRF's binding on 30S ribosome.

Page 5, 1208 and so on. The description of 30S modifications induced by the binding of Val-tRF suffer from the absence of a real "wet" control, using the same ribosomes and cryo-EM procedure. In other words, a map without Val-tRF is necessary here to check whether the h24, h44 and h45 helices are not altered. It could simply come from empty 30S particles that were not used in the final reconstructions.

Response: We appreciate the reviewer's highlighting of the importance of a real wet control. In

fact, the structure of EMD-34862 (Figure 4A) utilized 30S ribosomal subunits from the same ribosome preparation as those used to analyze the Val-tRF-30S complex. Additionally, during the electron microscopy data analysis of the Val-tRF-30S complex, no empty sub-classes lacking Val-tRF were identified.

Overall global resolutions are rather poor, which can be explained by the small amounts of particles and the dynamics of the 30S. However, what are the local resolutions of the Val-tRF described in the three complexes? If they are too low did the authors use rigid body fitting? What is the precision of their atomic models?

Response: The Val-tRNA residue 1-26 derived from previous crystal structure model were first docked in the EM map. The rigid body fitted model was subjected into phenix for real-space refinement. During Phenix cryo-EM real-space refinement, the atomic model is iteratively adjusted to improve its fit to the experimental cryo-EM density map. This process involves optimizing parameters such as atomic coordinates, atomic B-factors (temperature factors), and occupancy values to maximize agreement between the model and the experimental data. The refinement also considers the local resolution variation within the density map, allowing for more accurate fitting in regions of higher resolution. Figure 5A and B show the differences of initial rigid fitting and after real-space refinement.

Figure 5 Cryo-EM density showing the fitting of Val-tRF. (A) Initial rigid fitting of Val-tRNA fragment derived from crystal structures were fitted in the density. (B) After phenix real-space refinement, Val-tRF were fitted into the map.

Methods

Do the authors use montioncor2 (line 344) and then patch motioncorrection in cryosparc (line 350)? Please explain

Response: We apologize for any confusion regarding the image processing. The images were processed using cryoSPARC. The *.ERR movies were first imported into cryoSPARC and subjected to "patch motion correction." The resulting .mrc files were then utilized for CTF correction and particle picking. We have revised the relevant section in the methods.

Line 349 : Tilted data ? Please explain: did the authors used titled series?

Response: Tilt series data is collected using a transmission electron microscope (TEM) equipped with a cryo-holder capable of tilting the specimen. We used cryoSPARC to process the titled data including “Import Tilt Series” “Motion Correction”, “CTF Estimation” “Particle Picking” and “Volume Reconstruction”.

I am a bit confused by the particle numbers:

Starting from 854284 particles the authors keep only $9742 + 5281 + 6795 = 21818$ particles to reconstruct the three major classes. That means that 97.5% are not used? How can the authors claim that only 2.5% of the particles are biologically relevant, without overinterpretation?

Response: We appreciate the reviewer for highlighting the limited number of particles used to solve the three reported structures. Approximately 97.5% of the particles were utilized to solve around 30 additional structures without the precise adjustment of the Val-tRF models (as shown in Figure 3). As illustrated in Figure 3, the majority of Val-tRF binding patterns observed in the other ~30 structures closely resemble those found in structures I and II. Therefore, we consider structures I and II to be the most robust models, effectively explaining how Val-tRF influences translation.

March 11, 2024

RE: Life Science Alliance Manuscript #LSA-2023-02488-TR

Dr. Jie Zhou
Zhejiang University
Life Sciences Institute
866 Yuhangtang Rd.
Hangzhou, Zhejiang 310058
China

Dear Dr. Zhou,

Thank you for submitting your revised manuscript entitled "Structural basis of translation inhibition by a valine tRNA-derived fragment". We would be happy to publish your paper in Life Science Alliance pending final revisions necessary to meet our formatting guidelines.

- please be sure that the authorship listing and order is correct
- please upload your main and supplementary figures as single files
- please add a Running Title and a Summary Blurb/Alternate Abstract to our system
- please add ORCID ID for the secondary corresponding author -- they should have received instructions on how to do so
- please add a Category for your manuscript in our system
- please add the Twitter handle of your host institute/organization as well as your own or/and one of the authors in our system
- please remove your figures from the manuscript file
- please incorporate any points from the Conclusion section into the Discussion; we only allow a Discussion section
- please add an Author Contributions section to your main manuscript text
- please add your main, supplementary figure, and table legends to the main manuscript text after the references section
- please upload your Table in editable .doc or Excel format
- please add callouts for Figures 1A,B,D,E; 2G,H; 4C; S1A-C; S2A-C; S3A-E; S4A-L to your main manuscript text
- in the Data Availability section, please say where the structures have been deposited

A. FINAL FILES:

B. MANUSCRIPT ORGANIZATION AND FORMATTING:

Sincerely,

Reviewer #2 (Comments to the Authors (Required)):

The authors addressed all point raised during the first round of revision.

Minor comments: 1. Typo in supl figure 9: Poly-phe at the start of the sentence.

2. In the methods section of of the poly-phe assay somethimes pH and PH is used.

3. I would recommend introducing references in the methods part as well. These methods have been used before (maybe in different variations). References will help others to establish and modify their protocols.

Reviewer #3 (Comments to the Authors (Required)):

The authors have carefully and thoroughly gone through all my criticisms and have updated their work accordingly. I think this has significantly improved the transparency of the manuscript. I support its publication in LSA.

March 22, 2024

RE: Life Science Alliance Manuscript #LSA-2023-02488-TRR

Dr. Jie Zhou
Zhejiang University
Life Sciences Institute
866 Yuhangtang Rd.
Hangzhou, Zhejiang 310058
China

Dear Dr. Zhou,

Thank you for submitting your Research Article entitled "Structural basis of translation inhibition by a valine tRNA-derived fragment". It is a pleasure to let you know that your manuscript is now accepted for publication in Life Science Alliance. Congratulations on this interesting work.

DISTRIBUTION OF MATERIALS:

Again, congratulations on a very nice paper. I hope you found the review process to be constructive and are pleased with how the manuscript was handled editorially. We look forward to future exciting submissions from your lab.

Sincerely,
